



# Modeling the impact of river discharge and wind

## on the hypoxia off Yangtze Estuary

Jingjing Zheng[1,2], Shan Gao[1,3], Guimei Liu[1,3*], Hui Wang[1,3]

1. National Marine Environmental Forecasting Center, Beijing 100081, China

2. State Key Laboratory of Marine Environmental Science, Xiamen University, Xiamen 361005, China

3. Key Laboratory of Research on Marine Hazards Forecasting, National Marine Environmental Forecasting Center , Beijing 100081, China

**Abstract:** The phenomenon of low dissolved oxygen (known as hypoxia) in coastal ocean system is closely related to a combination of anthropogenic and natural factors. Marine hypoxic occurs in the Yangtze Estuary, China with high frequency and long persistence. It's known that it related primarily to organic and nutrient enrichment influenced by river discharges and physical factors, such as water mixing. In this paper, a three-dimensional hydrodynamic model was coupled to a biological model to simulate and analyze the ecological system of the East China Sea. By comparing with the observation data, the model results can reasonably capture the physical and biochemical dynamics of the Yangtze Estuary. In addition, the sensitive experiments were also used to examine the role of physical forcing (river discharge, wind speed, wind direction) in controlling hypoxia in waters adjacent to the Yangtze Estuary. The results showed that the wind field and river discharge have significant impact on the hypoxia off the Yangtze Estuary. The seasonal cycle of hypoxia was relatively insensitive to synoptic variability in the river discharge, but integrated hypoxic areas were sensitive to the whole magnitude of river discharge. Increasing the river discharge was shown to increase hypoxic areas, while decreasing the river discharge was tended to decrease hypoxic areas. And the variations of wind speed and direction had great impact on the seasonal variability of hypoxia and the integrated hypoxic areas.

**Key word:** Yangtze Estuary, wind, river discharge, hypoxia

Fundation item: The National Natural Science Foundation of China under contract No. 41222038, 41076011,41206023 ; from the National Basic Research Program of China ("973" Program) under contract No. 2011CB403606;and.the "Strategic Priority Research Program" of the Chinese Academy of Sciences Grant No. X DA1102010403.

*Corresponding author, E-mail: liugm@nmefc.gov.cn



## 1. Introduction

In recent decades, the eutrophication of water body driven by excess nutrient loads from lands due to human activities is increasing year by year, which leads to an enhancement of hypoxic zone (Murphy et al., 2011; Rabouille et al., 2008). Dissolved oxygen (DO) threatens marine animals when its concentration is lower than 2 mg/L or 62.5 umol/L, which is defined as hypoxia (Diaz, 2001; H Wei et al., 2007). When DO is less than 2.0 mg/L, the majority of marine aquatic organisms will be dead, especially benthic animal (Karlson et al., 2002). Hypoxia is one of the most severe environmental issues affecting estuarine and coastal marine ecosystems around the world. Hypoxia can reduce the diversity of marine species, change the community structure of marine organisms, reduce the richness of fish and benthic animals, and thus affect the fishery production and bring about direct or indirect economic loss (Yin et al., 2004).

The hypoxia off Yangtze Estuary was first found in 1959 (Gu, 1980). In recent years, with the increase of global warming and pollutant emissions, the hypoxic area off Yangtze Estuary expanded fast and became one of world's largest costal hypoxia (Vaquer-Sunyer and Duarte, 2008). In the 1950s, the occurrence probability on hypoxia off Yangtze Estuary in summer was 60%, while after 1990s hypoxia occurrence probability reached 90%, and hypoxic areas which were greater than 5000 km$^2$ basically occurred after the end of 1990s (Wang, 2009).

The formation of hypoxia adjacent to the Yangtze Estuary is a complex process, which is the result of both physical and biochemical processes. Previous research had shown that the formation and evolution of hypoxia were closely related to river discharge, Taiwan Warm Current, wind speed, wind direction, bottom topography, and the degradation of organic (Li et al., 2011; Ning et al., 2011; Wang, 2009; X Li, 2011; Zhou F, 2010; Zhu et al., 2011). It was generally accepted that the increase in hypoxic extent was mainly driven by the rising anthropogenic nutrient inputs from the Yangtze River. However many studies had indicated that the physical factors have important contributions to the formation of the annual variation of hypoxia. Wilson et al. (2008) found that the wind-driven circulation had an important influence on the West Long Island Strait stratification and vertical mixing. When the wind direction changed in summer, it prevented the exchange of dissolved oxygen from the bottom to surface, which made the West Long Island Strait appear hypoxia. Through numerical simulation, Obenour (2015) found that the river nutrient concentration was very important for the formation of hypoxia. However the stratification, which presented as a function of river discharge, wind speed and wind direction, contributed to a larger extent to the interannual variability in hypoxia. Due to the limitation of observation data, it was difficult to fully understand the temporal and spatial variation of hypoxia, and quantitatively described the influence of physical factors on the hypoxia off the Yangtze Estuary. In this paper, a three-dimensional hydrodynamic model (Regional Ocean Model System, ROMS) coupled nitrogen cycle model described by Fennel



et al. (2006) were used to simulate ecosystem of the East China Sea. Through this coupled model, the effects of
river discharge, wind speed and wind direction on dissolved oxygen at sublayer and bottom of sea waters off the
Yangtze Estuary were analyzed quantitatively.

**2. Model description**
**2.1 Physical model**
The physical circulation model used in this study was based on the Regional Ocean Model System, which
was a free-surface, terrain-following, primitive equations ocean model (Haidvogel et al., 2008; Shchepetkin A F,
2005). This model was built for the East China Sea (114-143$^{o}$E, 22.3-53.1$^{o}$N) with a 1/15 degree horizontal
resolution and 30 vertical layers (Fig. 1a). The models' terrain-following vertical layers were stretched to result in
increased resolution near the surface and bottom. A fourth-order horizontal advection scheme for tracers, a
third-order mixing by Mellor and Yamad were used in the model (Mellor G 1982). At the offshore open boundary,
we employed Chapman's condition for surface elevation (Chapman, 1985), Flather's condition for barotropic
velocity (Flather, 1976), and a combination of radiation condition and nudging for tracers (Marchesiello et al.,

2001).

The model was initialized with climatological temperature, salinity from the Generalized Digital
Environment Model (GDEM). The south, north and east of the pattern were open boundary, and the west was
closed boundary. On the open boundary, the average profile of temperature, salinity and sea level were derived
from Simple Ocean Data Assimilation (SODA). The model was forced with net heat flux, fresh water flux, short
wave radiation, wind stress and so on, which derived from NCEP/CFSR reanalysis. In the model, the runoff of the
Yangtze River was determined by the month average discharge of the Datong station (Liu Xincheng, 2002).
**2.2 Biological model**
The biological module was based on the nitrogen cycle model described in the study by Fennel et al. (2006).
The model included ten state variables: two species of dissolved inorganic nitrogen (nitrate, $NO_3$ and ammonium,
$NH_4$), one phytoplankton group, one zooplankton group, chlorophyll-a, two species detritus representing large,
fast-sinking particles and suspended, small particles, and oxygen, total inorganic carbon, alkalinity. The source of
dissolved oxygen in the model was air–sea gas exchange, primary production, and the sink included the
respiration zooplankton, nitrification, decomposing detritus and sediment oxygen consumption(Fennel et al.,
2013). At the sediment-water interface, the model assumed "instantaneous remineralization" by which organic
matter that sank to the bottom was remineralized immediately to ammonium and oxygen was taken up





immediately as well. The initial fields and boundary conditions for the biological tracers were derived from the
World Ocean Atlas 2009 (WOA2009). The initial value of ammonium was set to 1mmol/m$^3$.The chlorophyll-a
concentration was extrapolated in the vertical direction from surface values specified by SeaWiFS monthly
climatological data and the extrapolation was based on the generalized Gaussian curve of typical pigment profile
(Lewis et al., 1983; Morel and Berthon, 1989). The initial values of phytoplankton, zooplankton and detritus
concentrations were set to 0.5, 0.25 and 0.25 times of chlorophyll-a concentration, respectively. Ammonium can
be rapidly nitrified into nitrate at the inner estuaries. Therefore, only nitrate discharged from the Yangtze River
was considered in the model.

After the physical model had spun-up for 10 year since climatological 1st January, the coupled

physical-biological model was simulated from 1st January 2006 to 31th December 2010. A series of numerical
experiments in 2010 were set up to study the influence of river discharge, wind speed and wind direction on
hypoxia adjacent to the Yangtze Estuary.

**3. Results**
**3.1. Model validation**

The comparison of sea surface temperature (SST) and sea surface salinity (SSS) in August between model

and GDEM were shown in Fig. 2. Apparently, the model results of SST and SSS were similar to GDEM data. The
spatial distribution characteristic of SST was gradually increasing from north to south. The model results of the
SST along the coastal waters were higher than GDEM data, especially offshore of Bohai and the Yellow Sea. In
August, under the control of southerly wind, the Yangtze diluted water expanded to the northeast direction. From
the difference of SSS between simulated results and GDEM data, we found that a certain deviation between them
near the Yangtze Estuary. In reality, the GDEM data was not very accurate along the China coast, due to its low
spatial resolution. The distributions of SSS in our model were comparable to the results conducted by Liu et al.

(2010).

Fig. 3a,3b showed that the comparison of sea surface chlorophyll-a concentration between the model results

and SeaWiFS-derived data in August. It can be seen that the patterns of chlorophyll-a were comparable to the
SeaWiFS-derived data. For example, the distribution of chlorophyll-a was extended to northeast in summer. The
chlorophyll-a concentration was 5-10 mg/m$^3$ in surface waters adjacent to the Changjiang Estuary, then decreased
rapidly eastward to the open shelf, where values were 0.1–0.5 mg/m$^3$. The modeled chlorophyll-a values in the
inner and mid shelves were quite close to the SeaWiFS-derived data, while the model generally underestimated
the chlorophyll-a in the outer shelf. Compared with the SeaWiFS-derived data, the concentration of chlorophyll in




the south of 28$^{o}$N was also lower. This was mainly due to only consider the nutrient inputs form the Yangtze River
in the model.

In-situ nitrate and ammonium distributions along section 30$^{o}$N (see Fig. 3) in August were used to evaluate

the simulation capability of biological model. Both in-situ and model nitrate distributions showed that the
presence of relative high nitrate in the bottom (>4 mmol/m3), but low (< 2 mmol/m3) in the surface. An analogous
structure was present in the in-situ temperature and salinity distributions located between 123$^{o}$E and 124$^{o}$E
reported by Wang (2009), so that upwelled water inside the trough involved not only low temperature and high
salinity but also high nutrients. Form the Fig. 3c,3d, a definite deviation between the model and in situ data can be
seen. The high nitrate concentration appeared above water layers of 25 meter along the section of 123$^{o}$E, but the
model failed to reproduce it. The underestimation probably resulted from the insufficient dispersion of high nitrate,
which was source from Yangtze river diluted water. Along the section of 30$^{o}$N, the patterns of model ammonium
resembled that of observations. Ammonium was relatively high in the near-shore waters (>2 mmol/3), then
decreased toward the open waters (<1 mmol/m3). An obvious high-concentration area was located between 123$^{o}$E
and 124$^{o}$E, which reconfirmed that the Taiwan warm upwelling brought the bottom water with high nutrients to
the surface.

Fig. 4 showed the comparison between in-situ and model results of dissolved oxygen and nitrate

concentration at the stations (see Fig.1c). It can be seen that the simulated dissolved oxygen and nitrate resembled
the in-situ data. The root mean square error of RMS on surface dissolved oxygen, bottom dissolved oxygen,
surface nitrate and bottom nitrate were 0.55, 0.56, 1.58 and 1.94 respectively. The concentration of dissolved
oxygen in the surface layer was generally high (>6 mg/L), while was low in the bottom. The difference of nitrate
concentration between the model results and in-situ data was relatively obvious. The reason may be that the river
input of nitrate was monthly average in the model.

The distribution of dissolved oxygen adjacent to the Yangtze Estuary in September 2010 was showed in Fig.5.

On the whole, it presented that the distribution of dissolved oxygen was high in the north and low in the south. A
closed area with oxygen level less than 2.0 mg/L was appeared off Yangtze Estuary, and was extended along the
Yangtze Estuary to the Zhejiang coastal water. The model results of dissolved oxygen were similar to that
observed by Liu (2012).

Model validation results showed that the model can reproduce the variations of biological variables and the

hypoxia in summer at the bottom of the Yangtze Estuary in a certain extent. There were some deviations from the
observed results in some areas, which could due to the complex biogeochemical cycle, parameterization, physical
field and so on. At least, it's suggested that the model can capture some basic conditions of the key physical and



biological processes in the Yangtze Estuary, which is helpful for further exploring the mechanism of the formation
of hypoxia off the Yangtze Estuary.
**3.2 The impacts of river discharge on hypoxia**
To evaluate the role of the seasonal variation in river discharge on the hypoxic area, we conducted a
sensitivity experiment (denoted as 'Qconst') where river discharge was set to the annual mean value for 2010 and
the other conditions were the same as the Base model run (Table 1). The extent of hypoxic water off the Yangtze
Estuary can be quantified by calculating the total area of water that has bottom dissolved oxygen concentrations
below different threshold value (1.0 mg/L, 2.0 mg/L, 3.0 mg/L). If there was no special note, the hypoxic area was
denoted as the area of DO<2 mg/L in this paper. In the Base model run, hypoxia zone was first appeared off the
Yangtze Estuary in July, then reached its peak in August and September, finally it was reduced and gradually
disappeared in October (Fig.6a). The results were consistent with the findings of previous work conducted by Wei
(2015). As shown in Fig. 6a, the seasonal variations in hypoxic area in the Qconst run were almost consistent with
the Base model run, which suggested that the seasonal variation of hypoxic area was not remarkably affected by
the temporal variation of the river discharge of Yangtze River. In addition, the area of the bottom dissolved
oxygen concentration less than 1 mg/L, 2 mg/L and 3 mg/L decreased by 32%, 46%, 19% (Table 2) in the Qconst
run. Next, to evaluate the role of magnitude of river discharge, we conducted Q2 and Q0.5 sensitivity experiments
where the temporal variation in river discharge was preserved but the magnitude was doubled or halved,
respectively for 2010. Doubling the river discharge increased nearly 7 times of the area where the bottom
dissolved oxygen concentration was below the minimum threshold (1 mg/L). A reduction in the river discharge by
50% decreased the total integrated hypoxic area by 20% to nearly 63% under different threshold values.
Water stratification prevented the vertical exchange of dissolved oxygen, and it was an essential condition for
the formation of hypoxia. In this paper, Frequency Brunt-Vaisala ($N^2$) was calculated to quantify the stratification
strength, and it was denoted as the stratification strength between the surface and bottom water (Goni et al., 2006).
The greater the $N^2$, the stronger stratification of the water was. The calculating area of $N^2$ was showed in Fig. 1c
of the red rectangle.
$$N^2 = -\frac{g}{\rho}\frac{\partial \rho}{\partial z}$$
As shown in Fig. 6b, the stratification of water adjacent to the Yangtze Estuary had a significant relationship with
the river discharge. Doubling the river discharge increased markedly the stratification strength, while reducing the
discharge to 50% of the Base model run led to a significant decrease in the stratification. Fig. 7 showed the
correlation of concentration between the bottom dissolved oxygen and the $N^2$. We could found that $N^2$ was



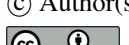

inversely proportional to the concentration of bottom dissolved oxygen. The stronger the stratification, the lower
concentration of bottom dissolved oxygen. The absolute values of correlation coefficient were higher than 0.6, and
the maximum of absolute correlation coefficient was 0.73 in the Q2 run.
**3.3 The impacts of wind forcing on hypoxia**
The wind forcing also had a significant effect on the occurrence and development of hypoxia. Increasing
wind speed enhanced the vertical mixing of water, and promoted the vertical exchange of dissolved oxygen, which
would lead to break the formation of hypoxia (Scully, 2010b). The Yangtze Estuary is mainly controlled by the
East Asian monsoon. The wind speed and wind direction have obvious seasonal variability over the Yangtze
Estuary, namely low magnitude and dominantly southerly direction in summer (July to September) and relatively
high magnitude and northerly direction in winter. In order to evaluate the role of seasonal changes in wind forcing,
we conducted wind runs where the August wind (denoted as 'WAug') and January wind (denoted as 'WJan') was
repeated each month of 2010, respectively. By forcing the model in this way, the winds had daily variations
associated with the passage of weather systems, but the seasonal variation of speed and direction were removed.
Although this kind of situation did not exist in practice, it was helpful for us to further understand the impacts of
wind forcing on hypoxia. The seasonal variability of wind speed and direction had a great impact on the seasonal
cycle of the simulated hypoxic area. When strong, northerly winds in January were repeated all year in the WJan
run, the hypoxia off the Yangtze Estuary was nearly disappeared. In contrast, when in the WAug run where light,
southerly wind conditions were repeated all year, the hypoxia was extensive from July to October (Fig. 10a), and
the integrated hypoxic area (DO<2 mg/L) increased roughly 25% compared to the Base model run (Table 2).
Hypoxic area also showed obvious seasonal changes. Hypoxic area was the largest in summer, while essentially
no hypoxia was simulated in the other months. Possible reason was that the phytoplankton was affected by water
temperature, and decreased in early spring and winter, so that the consumption of dissolved oxygen was reduced
due to organic matter decomposition.
To further examine the effects of wind speed on the hypoxia, we conducted a 'W0.9' wind run where the
wind speed decreased 10%, and a 'W1.1' wind run where the wind speed increased by 10% during the summer
(July to September). The magnitude of summer wind speed had a significant impact on the hypoxic area off the
Yangtze Estuary. When summer wind speed decreased, the integrated hypoxic area was increased by roughly 45%.
While increasing in summer wind speed led to a reduction by nearly 64% of hypoxic area (Table 2).
Previous study had suggested that hypoxic area was also sensitive to wind direction (Scully, 2010a; Xia and
Jiang, 2015). To further evaluate the influence of wind direction on the hypoxia, we conducted the model runs
where summer wind direction (July to September) were systematically varied in the model forcing. As is typical



in most summers, the wind during the 2010 summer is predominately from the south. In the wind direction
sensitivity runs, the modeled wind forcing during the summer (July to September) was rotated by 180 ° (W180 °),
negative 90 ° (W - 90 °), and positive 90 ° (W + 90 °), resulting in model forcing had the summer winds from the
north, east, and west, respectively (Table 1). Although these simulations were not realistic, they provide insights
into the impacts of wind direction on the hypoxia. When the summer wind direction was from the north (W180$^{\mathrm{o}}$),
the integrated hypoxic area off Yangtze Estuary was the minimum, and reduced by nearly 30% compared with
Base model run. The integrated hypoxic area was greatest when the summer wind came from the west (W + 90$^{\mathrm{o}}$),
and it was increased by nearly 20%. The summer wind direction from the east (W-90$^{\mathrm{o}}$) led to a reduction by nearly
10% of the integrated hypoxic area (Table 2).

**4. Discussion**
**4.1 River discharge**
Stratification was an important indicator of oxygen concentration in bottom water, which prevented the
exchange of dissolved oxygen from the surface to bottom, eventually resulting in hypoxia in the bottom water
(Rabouille et al., 2008). The simulated seasonal cycle of hypoxic area on the Qconst run was similar to the Base
model run (Fig. 6a), which suggested that the temporal variation in river discharge was not an important factor
controlling the seasonal cycle of hypoxic area. But the magnitude of Yangtze River discharge variation could lead
to significant changes on the hypoxic area. Increasing river discharge led to an expansion of the lighter, fresher
river plume water offshore and an enhancement of stratification (Fig. 8a), which limited the effective supplement
of surface high dissolved oxygen. This result consequently caused a reduction of bottom dissolved oxygen
(Fig.8c). And the integrated hypoxic area was increased by 92%, the lowest value of bottom dissolved oxygen
from the Base model run of 1.11 mg/L, reduced to 0.78 mg/L. Whereas decreasing river discharge reduced the
stratification, and thereby significantly increased the vertical oxygen flux through the pycnocline. As a result, the
integrated hypoxic area decreased by 55% and the lowest value of bottom dissolved oxygen increased to 1.32
mg/L. This findings were consistent with the results of previous work conducted by Scully (2013). In addition,
there was a significant negative correlation between the bottom dissolved oxygen and the stratification (Fig. 7).
When the water stratification was strong, the bottom dissolved oxygen was low. While the water stratification was
weak, and the bottom dissolved oxygen was high. Fig. 8a further showed that doubling or halving the river
discharge respectively enhanced or reduced stratification over the majority of the shelf in summer months,
resulting in decrease or increase the bottom dissolved oxygen.
Increasing river discharge leaded to offshore extension of the fresher river plume water (Fig.9b), bringing



rich nutrients to the eastward. As a result, surface chlorophyll over the majority of the eastern shelf was increased
in Q2 run relative to the Base model run (Fig.8b), which caused the continuous decrease in oxygen level at the
lower layer through decomposition of organic matters. In contrast, decreasing the river discharge confined the
river plume water to near the river mouths (Fig.9c), which limited the nutrients to eastward. As seen in Fig 8b,
surface chlorophyll was decreased over the majority of the shelf in Q0.5 run compared to the Base model run.
Thus, bottom dissolved oxygen was increased and hypoxic zone was decreased, due to the decomposition of
organic matters was reduced in the bottom water.
**4.2 Wind Forcing**

Strong wind could trigger strong vertical mixing and promote the vertical exchange of dissolved oxygen,

which broke the formation of hypoxia (Chen et al., 2014; Ni et al., 2014). Our simulated results showed that the
variation in wind speed and direction significantly influence the stratification and hence the hypoxic area. In the
WJan wind run, the strong wind homogenized the water column, reducing stratification and producing essentially
no hypoxia throughout the year. Fig. 11a showed that $N^2$ was negative in the most areas, except in the eastern part
of the Hangzhou bay, which suggested that the water mixing was strong and the vertical flux of dissolved oxygen
increased in WJan wind run. As a result, the bottom dissolved oxygen concentration was increased (Fig. 11c), and
there was almost no hypoxia in the bottom water off the Yangtze Estuary in WJan wind run. In addition, the
strong northwest wind in January resulted in strong estuarine residual velocities that brought the high dissolved
oxygen from the north of Yellow Sea to Yangtze River Estuary. Therefore, even though the average stratification
strength in WJan wind run was stronger than the Q0.5 run (Table 2), WJan nearly did not develop any hypoxia
throughout the year. From the Fig.11b, it could be seen that the surface chlorophyll concentration of WJan wind
run was significantly lower than the Base model run. The possible reason was that the strong wind from northwest
in WJan wind run drove downwelling due to Ekman dynamics, which made the concentration of surface
chlorophyll decreased, and ultimately led to an enhancement of dissolved oxygen through lower decomposition of
organic matters. These finds were the same as the results reported in the study conducted in the northern Gulf of
Mexico shelf by Feng (2014).Above all the reasons, in the WJan wind run there was almost no hypoxia in the
summer adjacent the Yangtze River Estuary.

In contrast, in the WAug wind run, the persistently weak wind enhanced the stratification and reduced the

vertical flux of dissolved oxygen, resulting in decreasing bottom dissolved oxygen and promoting widespread
hypoxia. In addition, the concentration of surface chlorophyll was increased adjacent to Yangtze Estuary and
along the coast of Zhejiang in the WAug wind run (Fig. 11b). This result suggested that bottom dissolved oxygen
declined with the decomposition of organic matters.





Simulations of hypoxic area showed significant variability in the response to wind speed. In the W1.1 wind
run, the increase in wind speeds was thought to play a key role in breaking down stratification and increasing the
vertical flux of dissolved oxygen. Increased wind speeds generally raised bottom dissolved oxygen concentration
(Fig. 11c). This finding was similar to the results conducted in the Louisiana Coasts, where the authors suggested
that wind-induced vertical mixing could result in significant reductions in the hypoxic area (Wiseman et al., 1997).
As shown in Fig. 11b, in the area of 28-32$^{o}$N 122-123.5$^{o}$E, decreased surface chlorophyll resulted in lower
dissolved oxygen consumption, which increased the dissolved oxygen in the bottom water. In contrast, in the
W0.9 wind run, the dissolved oxygen flux from the upper layers due to stronger stratification was reduced. And
increased surface chlorophyll concentration on the north of 30$^{o}$ led to huge dissolved oxygen consumption caused
by the decay of dead phytoplankton (Fig,11b). These results worked together to decrease the bottom dissolved
oxygen and create hypoxic expansion. Scully (2013) illustrated how changes in wind associated with stratification
and thereby hypoxia on the shelf. They found that increased wind speed decreased the hypoxic area, whereas
decreased wind speed facilitated hypoxia development. These findings were consistent with our results.
Changes in wind direction also significantly influence the hypoxic area off Yangtze Estuary. As seen in Fig.
12a, when the mean summer wind direction was from north (W180$^{o}$), stratification was decreased in the north of
31$^{o}$N and along the coast of Zhejiang relative to the Base model run. This would lead to an enhancement of
vertical dissolved oxygen flux. Whereas near to the Hangzhou bay, the vertical dissolved oxygen flux was reduced
associated with increased water stratification, resulting in decreased bottom dissolved oxygen (Fig. 12c).Under the
control of the northerly wind (W180$^{o}$), Zhejiang coastal current was enhanced, bringing the high dissolved oxygen
from the north of Yellow Sea to the Yangtze Estuary. This further led to increased bottom dissolved oxygen off
Yangtze Estuary. From the Fig. 12b, it can be seen that surface chlorophyll concentration in the W180$^{o}$ was lower
than the Base model run. We attributed this to the fact that wind from the north drive downwelling due to Ekman
dynamics. Decreased dissolved oxygen consumption was caused by the reduced decay of dead phytoplankton.
When the summer wind was from the East (W-90$^{o}$), the average water stratification was similar to the W180$^{o}$
wind run, but the integrated hypoxic area in W-90$^{o}$ wind run was larger than the W180$^{o}$. This may be related to
the horizontal distribution of chlorophyll concentration. As shown in Fig. 12b, surface chlorophyll concentration
was higher in W-90$^{o}$ wind run than the W180$^{o}$ run, which would lead to more dissolved oxygen consumption
caused by organic matter decomposition in W-90$^{o}$. When the summer wind was from the west (W+90$^{o}$),
stratification was the strongest in the three wind direction runs. And integrated hypoxic area also reached the
maximum. There can be three reasons. For the first reason, Fig. 12a showed that stratification was stronger in the
most areas, except in the north of the 31.5$^{o}$N relative to the Base model run, which suggested that in most areas





the water mixing was week and the vertical flux of dissolved oxygen reduced. For the second reason, under the
influence of the westerly wind, the residual flow would transport the low dissolved oxygen from the trough off the
Yangtze Estuary to the eastward. For the third reason, from Fig. 12b, it can be found an area of relatively high
chlorophyll concentration off Yangtze Estuary in W+90$^{o}$ wind run. This may be due to Yangtze River which was
rich in nutrients expand to the eastward under the controlling of westerly wind. As a result, it would promote the
growth of phytoplankton. The decomposition of dead phytoplankton was another important factor for the
decreased bottom dissolved oxygen. These reasons work together to create hypoxic conditions.

**5. Conclusions**

In this study, a three-dimensional coupled physical-biological model was used to analyze the hypoxia off to

the Yangtze Estuary. This study highlighted that river discharge, wind speed and wind direction all had significant
impacts on the hypoxia. The seasonal cycle of hypoxia was relatively insensitive to the temporal variability in
river discharge. But the integrated hypoxic area was very sensitive to the magnitude of river discharge. Increasing
in the magnitude of river discharge led to enhance stratification, promote the growth of phytoplankton associated
with higher nutrients, and thereby greatly increased hypoxic area off the Yangtze Estuary. In contrast, decreased
in the magnitude of river discharge reduced the stratification and surface chlorophyll concentration and hence
significantly decreased the hypoxic area.

Model simulations demonstrated that wind speed and wind direction not only play an important role in the

seasonal cycle of hypoxia, but also in the integrated the hypoxic area. When the winds in January were repeated
all year, the hypoxic zone was nearly disappeared as the result of the strong water mixing induced by strong,
northerly winds. While persistently weak winds from August enhanced stratification and facilitated hypoxia
development. Increasing wind speed weakened stratification and chlorophyll-a concentration, hence decreased the
hypoxic area, while decreasing wind speed did the opposite. Wind direction significantly influenced the extent of
hypoxia. Among the directions runs, the integrated hypoxic area was greatest when the summer wind came from
the west (W + 90$^{o}$), which was enhanced by nearly 20%. When the summer wind direction was from the north
(W180$^{o}$), the integrated hypoxic area off Yangtze Estuary was minimum. The integrated hypoxic area reduced by
nearly 10% when the wind was from the east (W-90$^{o}$).

The model did not include inorganic phosphorus and therefore assumed that primary production was limited

by light and nitrogen only. In future studies, the model needs to consider the dynamics of inorganic phosphorus. In
addition, increasing the number of monitoring cruises per year and setting up additional long-term moorings in the





East China Sea would be useful to further validate biogeochemical variables and environmental factors.

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





**Table.1    Model sensitivity experiment**

| Experiments | Description |
|---|---|
| Base model run | |
| Base | Base run with realistic river discharge and wind forcing in year 2010 |
| River discharge runs | |
| Qconst | River discharge was set to annual average value |
| Q2 | Double the river discharge |
| Q0.5 | Halve the river discharge |
| Wind runs | |
| WJan | Winds from January were repeated for every month of the year |
| WAug | Winds from August were repeated for every month of the year |
| W0.9 | Summer (July–September) wind magnitude was decreased by 10% |
| W1.1 | Summer (July–September) wind magnitude was increased by 10% |
| W180° | Summer (July–September) wind direction was rotated 180° |
| W-90° | Summer (July–September) wind direction was rotated negative 90° |
| W+90° | Summer (July–September) wind direction was rotated positive 90° |

Table.2   The total area of hypoxia area under different thresholds of DO,
and the average $N^2$ in summer and the year

| Experiments | Integrated Hypoxic Area ($10^3$ km$^2$ days) | | | Average $N^2$ | |
|---|---|---|---|---|---|
| | <1mg/L | <2mg/L | <3mg/L | Summer average | Yearly average |
| Base model run | | | | | |
| Base | 11.7 | 330.5 | 1163 | $5.58 \times 10^{-4}$ | $2.63 \times 10^{-4}$ |
| River runs | | | | | |
| Qconst | 8.0 (- 32%) | 179.6 (- 46%) | 939.3 (- 19%) | $3.80 \times 10^{-4}$ (-32%) | $2.24 \times 10^{-4}$ (-15%) |
| Q2 | 94.1(+704%) | 634.8(+92% ) | 1640.4(+41%) | $10 \times 10^{-4}$ (+79%) | $5.03 \times 10^{-4}$ (+91%) |
| Q0.5 | 4.33 (- 63%) | 147.1 (- 55%) | 927.7 (- 20%) | $3.28 \times 10^{-4}$ (-41%) | $1.59 \times 10^{-4}$ (-40%) |
| Wind runs | | | | | |
| WJan | 0 (-100%) | 13.9 (-96%) | 256.8 (- 78%) | $4.06 \times 10^{-4}$ (- 27%) | $1.90 \times 10^{-4}$ (-28%) |
| WAug | 25.4(+117%) | 412.8 (+25%) | 1435.3(+23%) | $6.25 \times 10^{-4}$ (+12%) | $3.02 \times 10^{-4}$ (+15%) |
| W0.9 | 49.0(+319%) | 478.0 (+45%) | 1296.8(+12%) | $6.29 \times 10^{-4}$ (+13%) | $2.80 \times 10^{-4}$ (+6%) |
| W1.1 | 0 (-100%) | 118.9 (- 64%) | 894.4 (- 23%) | $4.89 \times 10^{-4}$ (- 12%) | $2.45 \times 10^{-4}$ (- 7%) |
| W180° | 13.8 (+18%) | 235.5 (- 29%) | 909.4 (- 22%) | $4.97 \times 10^{-4}$ (-11%) | $2.45 \times 10^{-4}$ (- 7%) |
| W-90° | 14.6 (+25%) | 296 (- 10%) | 1020.5(-12%) | $5.07 \times 10^{-4}$ (- 9%) | $2.51 \times 10^{-4}$ (- 5%) |
| W+90° | 20.3 (+74%) | 390.9 (+18%) | 1240 (+7%) | $6.68 \times 10^{-4}$ (+20%) | $2.85 \times 10^{-4}$ (+8%) |






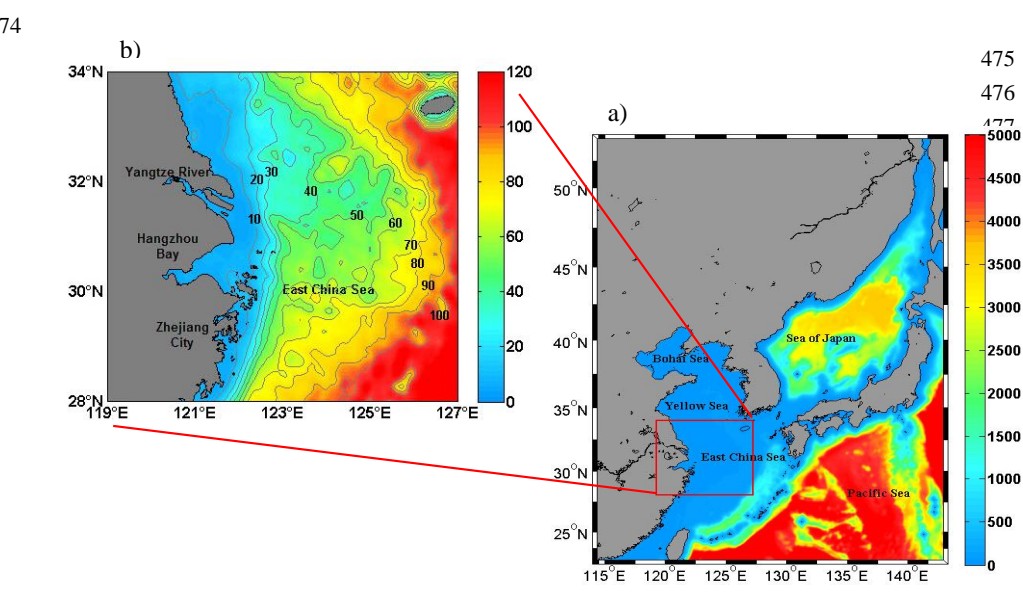



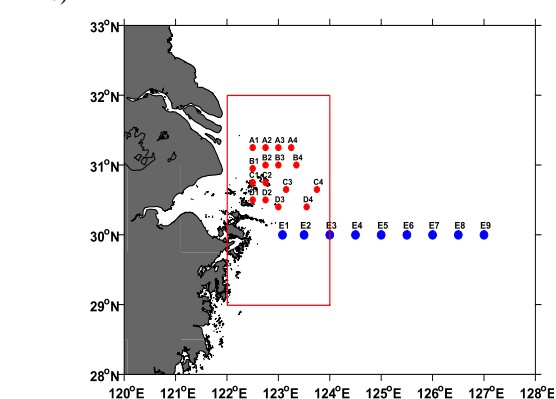

Fig.1   (a) Model domian and depth. Red box is the region of Fig.1b. (b) The model bathymetry (shading map, unit: meter) of East

China Sea. (c) Red dots are the station observation in August 2011, blue dots are the section observation in August 2011, the

red rectangle indicates the region used for the calculation of $N^2$.


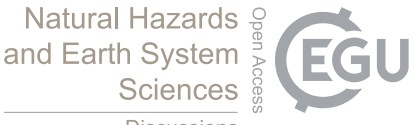

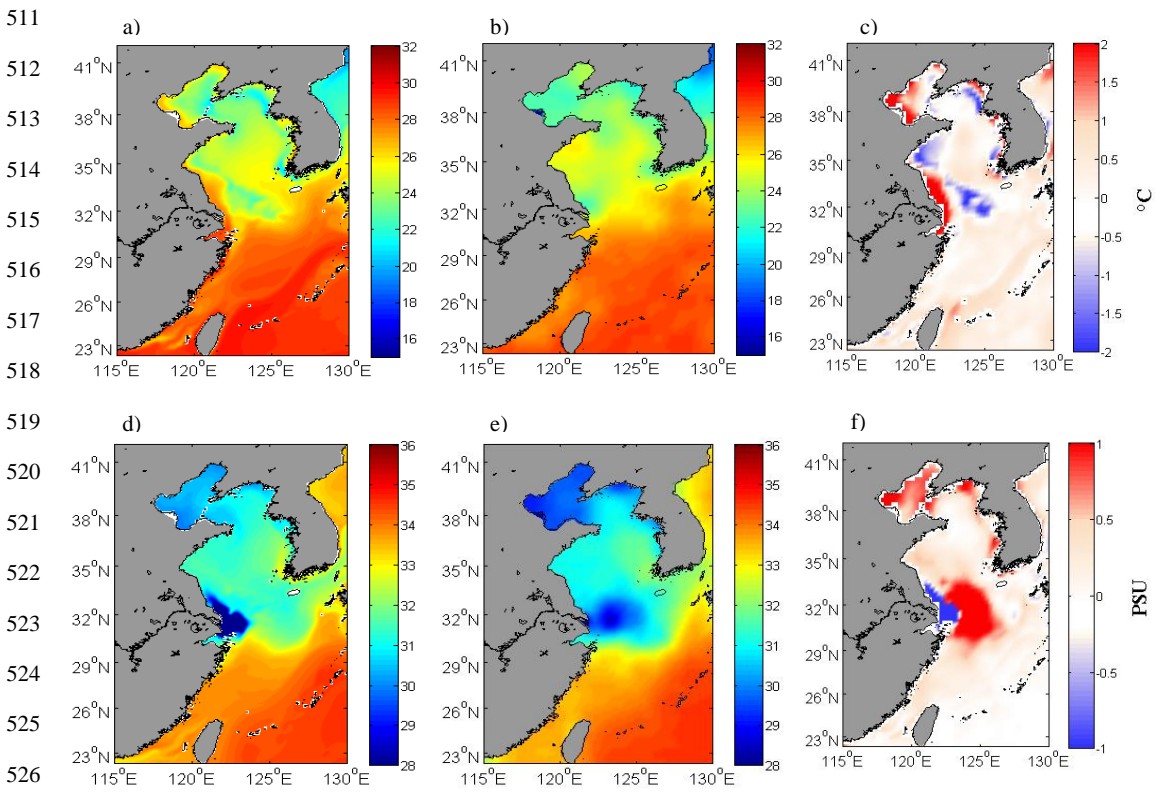

Fig.2    The comparison between modeled results (a) and GDEM data (b) of climatological field of SST in August, and the difference between them (c). The comparison between modeled results (d) and GDEM data (e) of climatological field of SSS in August, and the difference between them (f).

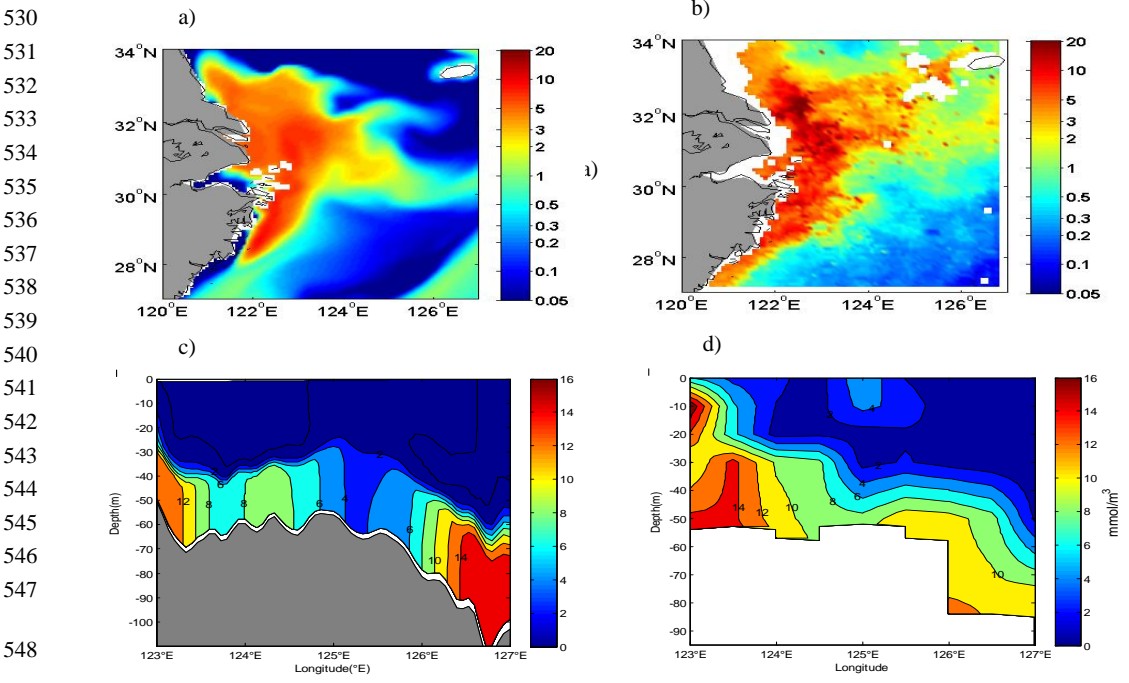





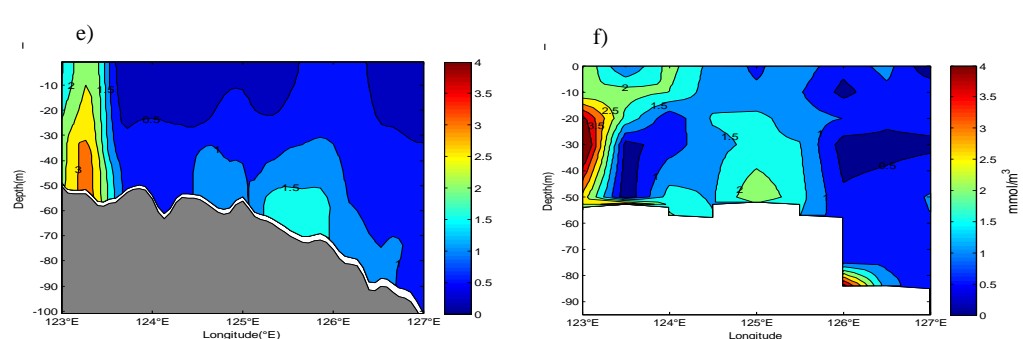

Fig.3 The comparison between modeled results (a) and SeaWiFS-derived data (b) of chlorophyll-a concentration in August 2010.

The comparison between modeled results (c) and in-situ data (d) of $NO_3$ along the section of 30°N in August 2011. The

comparison between modeled results (e) and in-situ data (f) of $NH_4$ along the section of 30°N in August 2011.

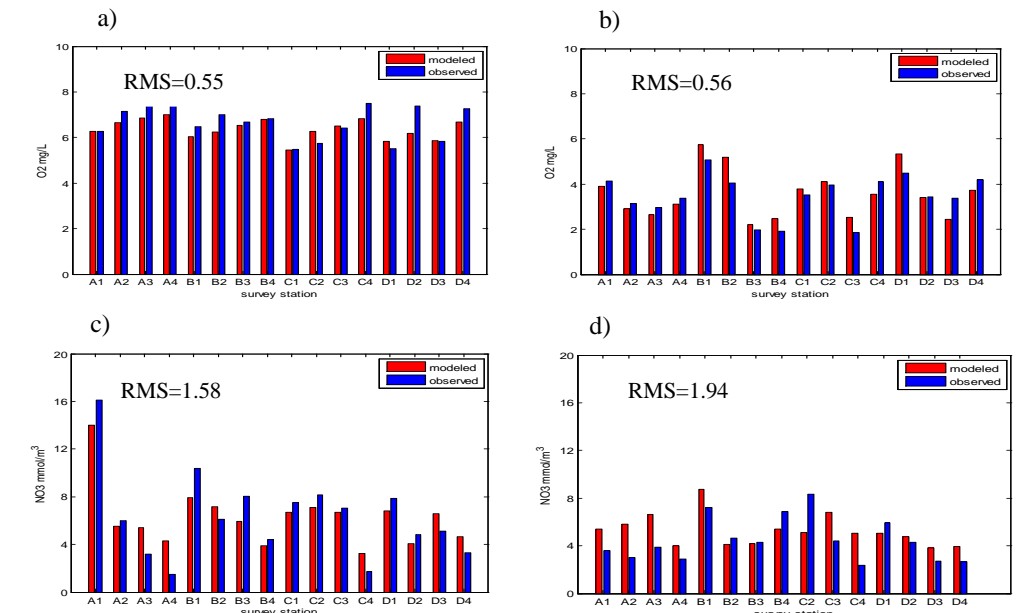

Fig.4 The comparison between modeled results (red bars) and observed data (blue bars) of $NO_3$ and DO at the stations in August

2011 (see Figure 1c). (a), (b) Respectively represent the surface DO and bottom DO; (c), (d) Respectively, represent the surface $NO_3$

concentration and bottom $NO_3$ concentration






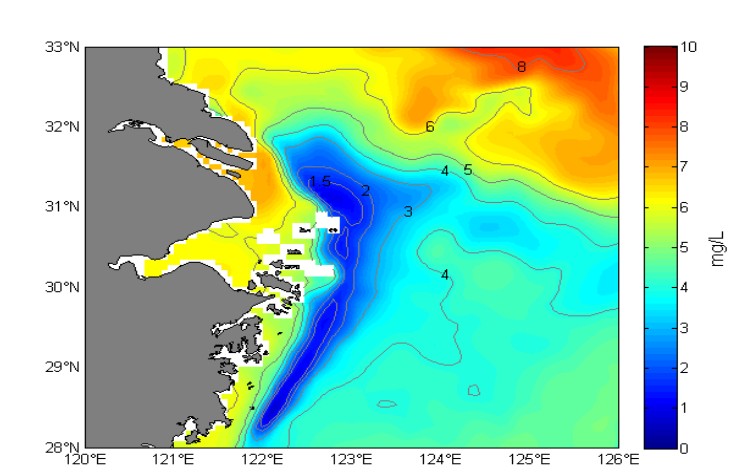

Fig.5    The modeled patterns of hypoxia zone off the Yangtze Estuary in September 2010

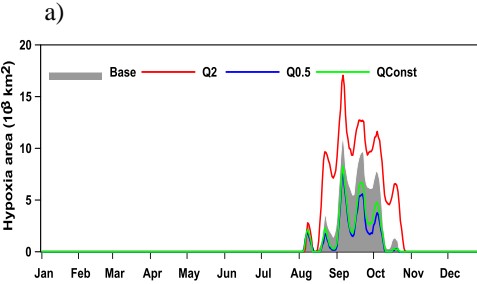 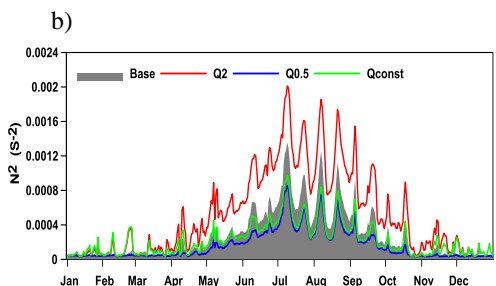

Fig.6    (a), (b) are the hypoxia area (DO<2mg/L) and $N^2$ in the river discharge experiment, respectively. The gray shaded part
represents the Base run and the red, blue, green curve represent Q2, Q0.5, Qconst, respectively.









Fig.7    The correlation between the bottom dissolved oxygen concentration and N$^2$ in the station of 123$^o$E, 31$^o$N. The blue curve represents the bottom dissolved oxygen concentration, and the red curve represents N$^2$. (a), (b), (c), (d) respectively represent the Base run, Qconst, Q2, Q0.5.





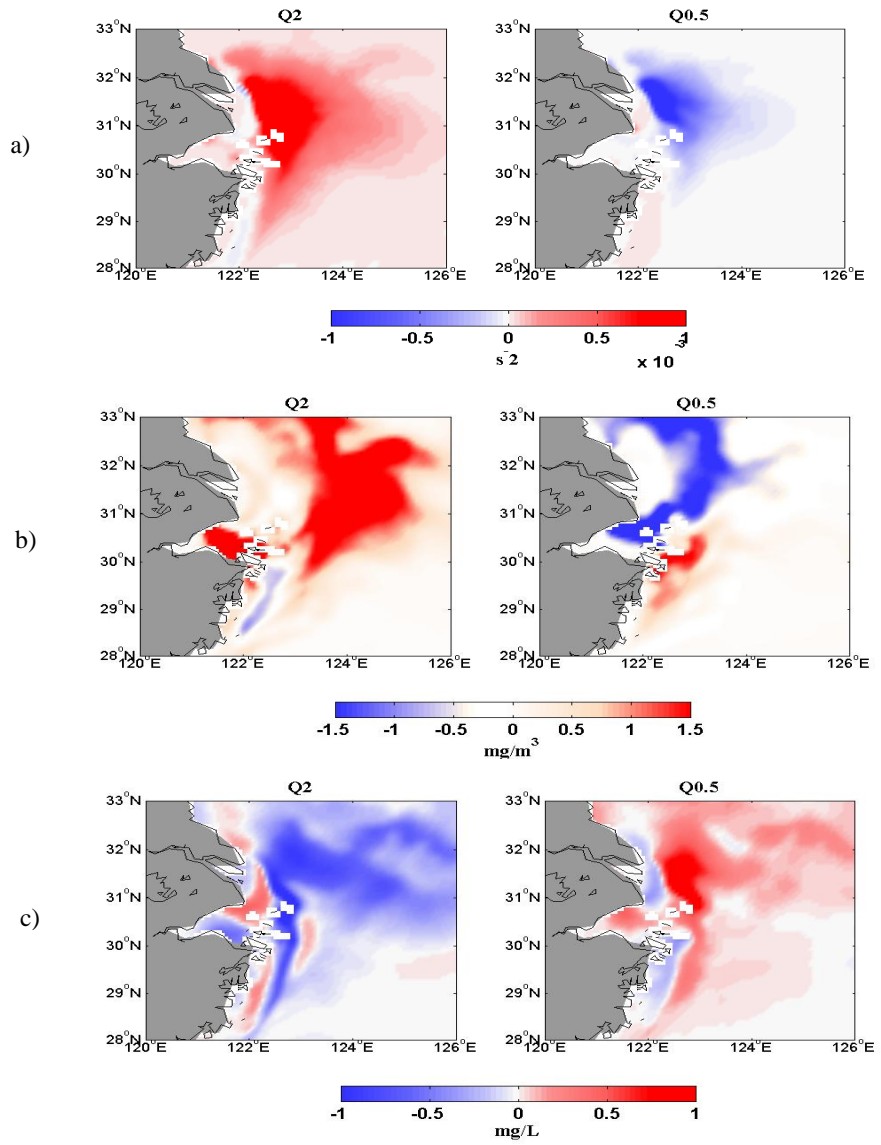

Fig.8    (a) The deviation in summer (7-9 months) stratification($N^2$) of Q2, Q0.5 from the Base run. (b)The deviation in summer (7-9 months) chlorophyll concentration of Q2, Q0.5 from the Base run. (c) The deviation in summer (7-9 months) bottom dissolved oxygen concentration of Q2, Q0.5 from the Base run.





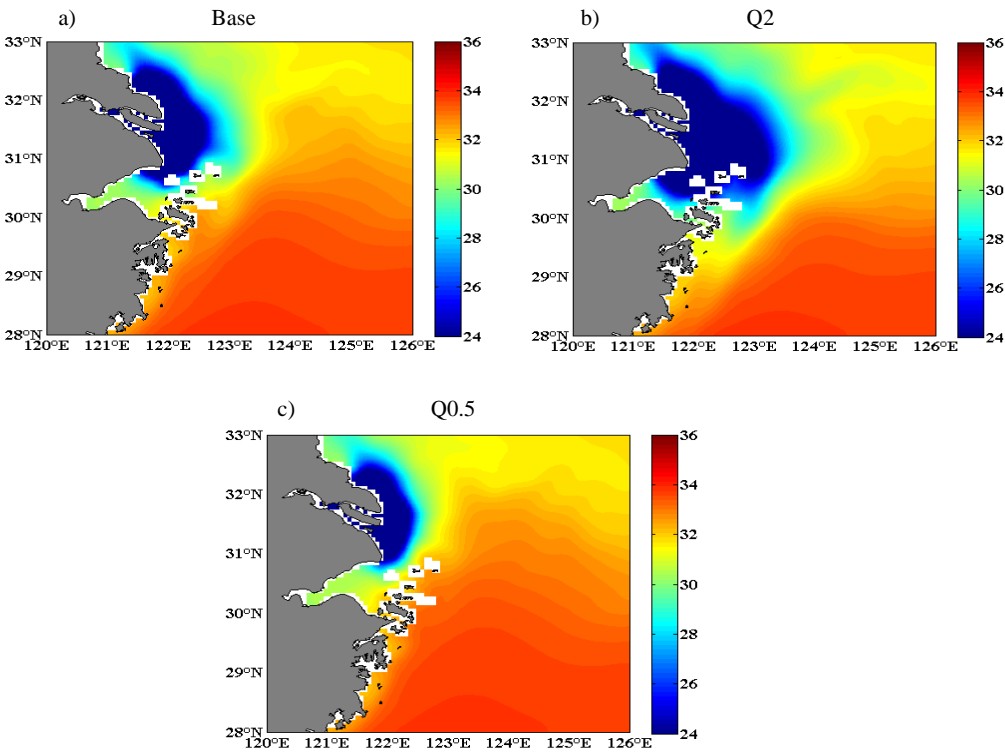

Fig.9 Averaged surface salinity for the period July to September in river runs. (a), (b), (c) respectively represent the Base run, Q2, Q0.5.





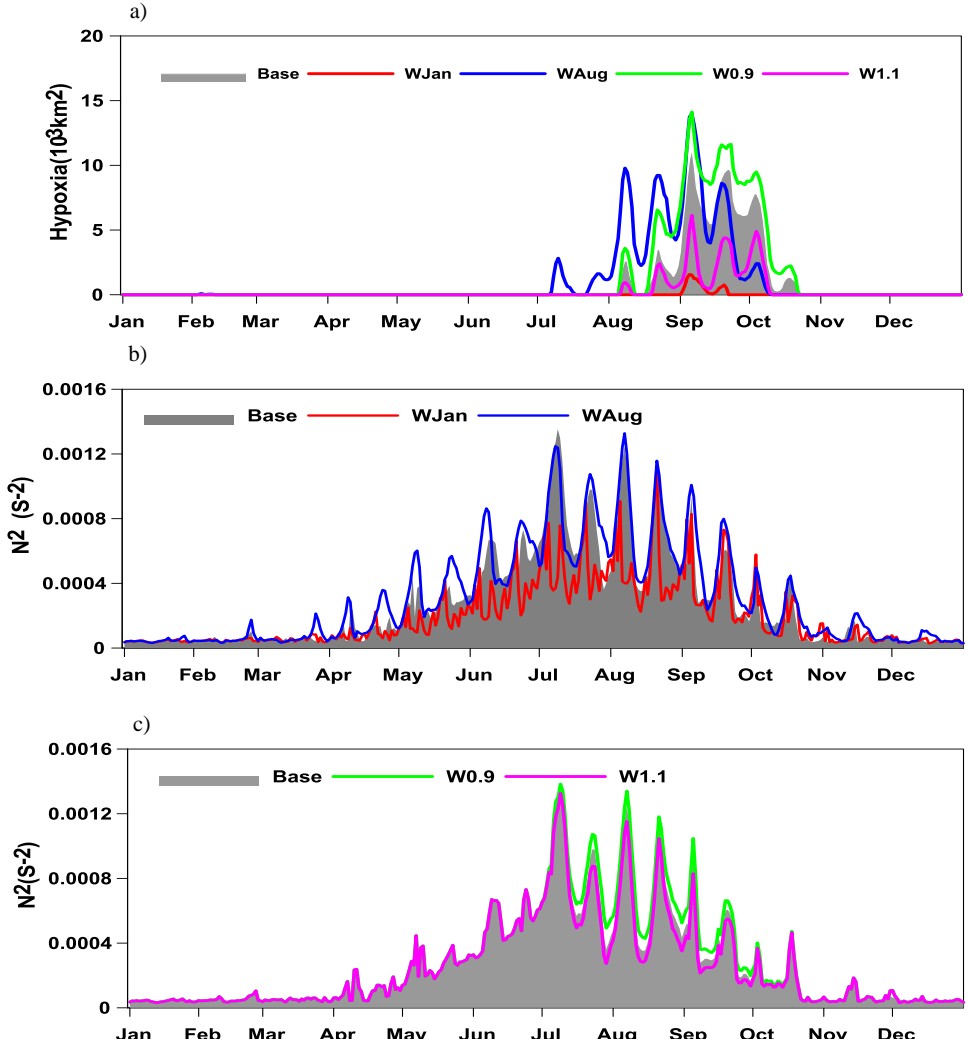

Fig.10    (a) The comparison of the simulated hypoxic area (DO<2mg/L) on the wind variation runs. (b),(c) Averaged stratification $N^2$

for the Base run (gray shadow) and wind variation runs



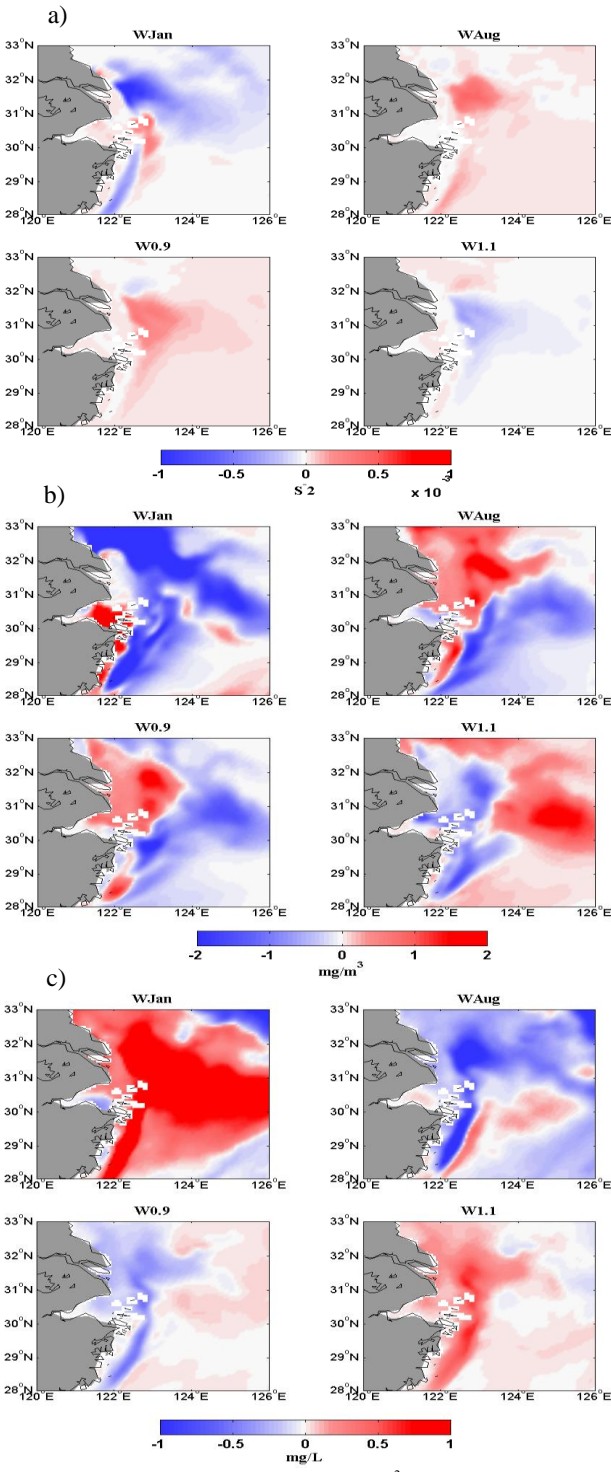

Fig.11  (a) The deviation in summer (7-9 months) stratification(N$^2$) of wind variation runs from the Base model. (b) The deviation in summer (7-9 months) chlorophyll concentration of wind variation runs from the Base model. (c) The deviation in summer (7-9 months) bottom dissolved oxygen concentration of wind variation runs from the Base model





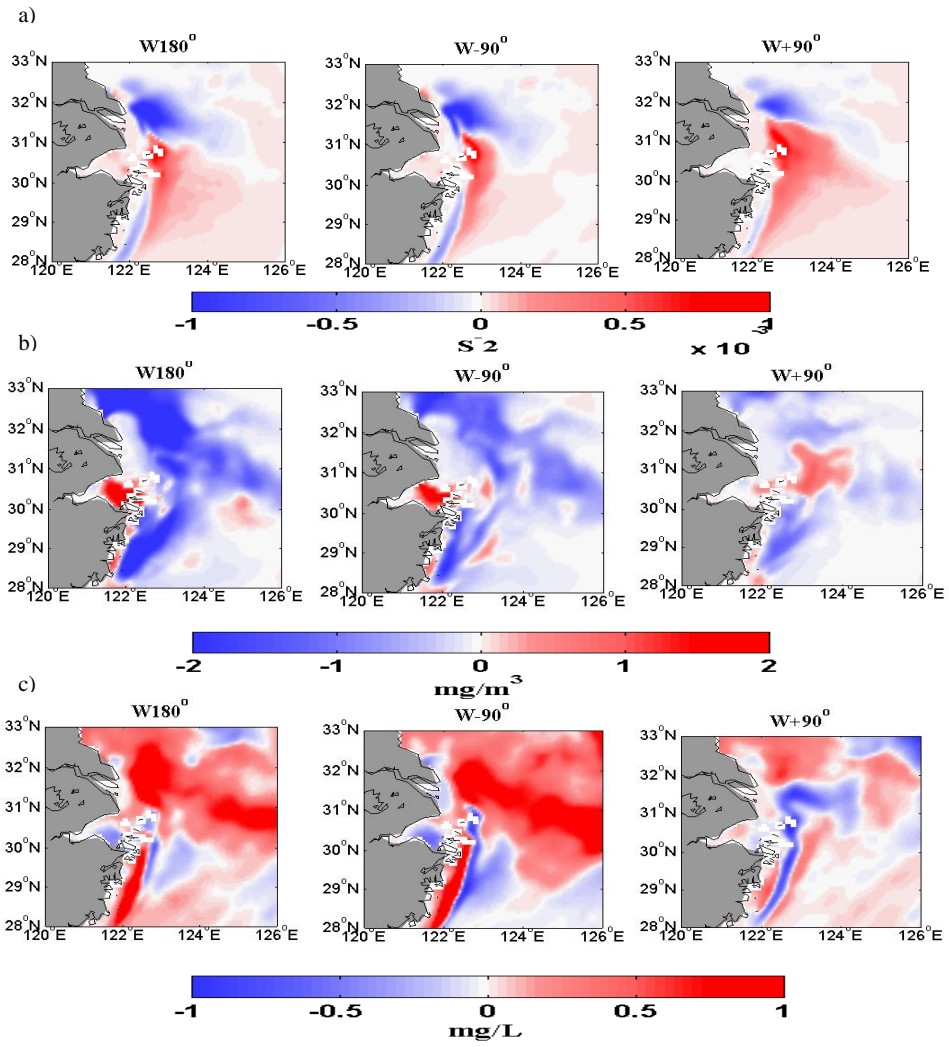

Fig.12 (a) The deviation in summer (7-9 months) stratification($N^2$) of wind direction variation runs from the Base model. (b) The deviation in summer (7-9 months) chlorophyll concentration of wind direction variation runs from the Base model. (c) The deviation in summer (7-9 months) bottom dissolved oxygen concentration of wind direction variation runs from the Base model