# Peer review of "Modeling the impact of river discharge and wind"

_Natural Hazards and Earth System Sciences, 2016_

## Referee Comment (RC1) · Anonymous Referee #1 · 20 Jun 2016

Review of "Modeling the impact of river discharge and wind on the hypoxia off Yangtze Estuary". Author(s): Jingjing Zheng, Shan Gao, Guimei Liu, Hui Wang

MS No.: nhess-2016-129 - MS Type: Research article

Special Issue: Situational sea awareness technologies for maritime safety and marine environment protection

The present paper compared simulated and observed data of the East China Sea. The study aims to demonstrate the compatibility of the three-dimensional coupled physical-biological model used with the observed data. Both physical and biological variables are studied. In particular nitrates and ammonia are analyzed all along the water column. Particular attention is given to the Yangtze Estuary. The authors demonstrate that the river discharge and the wind speed and direction play an important role on hypoxia events.

With reference to the criteria recommended by the Editor, and expressing an overall judgment, this work is surely to be considered as a good work for its Scientific Significance. It is of great interest regarding the comparison between model outputs and observations. Furthermore, it represents a serious contribution to the understanding of anoxia process in coastal marine areas.

Regarding the Presentation Quality, the judgment is to be considered "fair". In my opinion, the manuscript has to be revised in all the sections and in particular in the results section. Results and discussion sections are not very clear and well structured. Figures are not presented in sequential order and some figures are not explained at all in the results. Moreover, the time period of the observed and model data is reported in a confusing way and is sometimes omitted.

Finally, in my opinion this manuscript can be published after major revision.

Specific comments:

All figures should be explained in the results section. Some figures are not at all explained in the results section (Fig. 8, 9, 11 and 12) (from line 234). Please explain all the figures in the results section.

Furthermore figure should me recalled sequentially through the text. To avoid possible confusions, all the figures should be explained sequentially throughout the text. For example, Fig. 1 is cited for the first time after Fig. 2, 3 and 4 (line 140).

Some methodological aspects need to be clarified, mainly about the data used in the study. I suggest to add a section where the authors explain clearly the in situ data used in the study. Figure 1 should be explained in this section. In particular the time period (months and years) of the data used need to be clarified, both in the text and in the figures.

Fig. 2: "Comparison between model results and GDEM data in August". Are these data related to August 2010 or 2011? This part is not explained even in the text (line 108 – 112). Please clarify this part.

Fig.3 presents a comparison between model results and Sea-WiFS-derived data of August 2010, while nutrients ($NO_3$ and $NH_4$) data are referred to August 2011. Why? Chlorophyll is strongly related to nutrients availability, so the authors should explain this point. Please specify the period of the data used also in the text (in the results or add a section concerning the data sets used).

Line 117-118: only the month (August) is specified for chlorophyll data.

Line 126-127: only the month is specified for in situ-nutrients.

Fig.4: shows a comparison between model and observed data of $NO_3$ and dissolved oxygen (DO) in August 2011. While, in fig. 5 the distribution on DO is shown for September 2010. Why? If the observed data are referred to August 2011 what is the point to simulate September 2010? Maybe the authors want to relate the DO simulations to chlorophyll-a data (Fig.3), but this part is not clear. At least the authors should show the same period used for chlorophyll-a

(August 2010) and explain it in the text.

Fig. 7 shows the correlation between the bottom dissolved oxygen and the Brunt-Vaisala frequency ($N^2$). Are both variable calculated for 2010? Please specify it in the text.

Line 178: the authors recall Fig. 1c, however in this figure the authors affirm that the data were collected in the station during August 2011. What about Brunt-Vaisala frequency ($N^2$)? "…the red rectangle indicates the region used for the calculation of $N^2$" Is $N^2$ calculated for 2010 or 2011?

Figure 10a (line 202) is explained before figure 8 and 9. Please number the figures following a sequential order.

In figure 9 is shown the simulated surface salinity for the period July-September. Please specify the year.

The statistical treatment of the data should be improved.

Paragraph 3.1 model validation. Simulated surface temperature and salinity are compared with GDEM data, and differences between the two fields are shown in Fig.2. However, no statistical analyses have been conduced to demonstrate that the differences between simulations and observed data are not significant. Line 109: "Apparently, the model results SST and SSS were similar to GDEM data." To validate the model statistical analysis are necessary and this should be explain in the text.

Line 118-119: "It can be seen that the patterns of chlorophyll-a were comparable to the SeaWifs-derived data". The authors should add statistical analysis in order to compare the simulated and observed data (fig.3).

Line 141-142: the authors should calculate also BIAS and add it to figure 4.

Line 183-187: please specify the correlation coefficient used and add the p value.

Line144-145: this sentence should be moved to the discussion section

Line 149-150: Comparison with other studies should be expanded and moved to the discussion section.

Line 163-165: looking at fig. 6a, in the Base model run hypoxia zone appeared in August; and it disappeared in November.

---

## Author Comment (AC1) · 17 Jul 2016

The comment was uploaded in the form of a supplement:
http://www.nat-hazards-earth-syst-sci-discuss.net/nhess-2016-129/nhess-2016-129-AC1-supplement.zip

---

## Referee Comment (RC2) · Anonymous Referee #2 · 1 Aug 2016

This manuscript show a mechanism of the hypoxia off Yangze estuary, that the variation of wind is the main cause of seasonal varialility of hypoxia, while the river discharge can modify its magnitude. But the Fig. 10, it office the evidence that the contant wind (wind in August) can reproduce seasonal variation of hypoxia, although which is a month front to the base run. So I think the author shall make their conclusion with more confident evidence.

Major comments: 1. As mention above the Fig. 10a show the wind variation is not the role to make seasonal variation of hypoxia, which is conflicted with the main conclusion (L21-22). 2. Changing the wind direction will need another circulation pattern, which is not covered in this MS. So please do it carefully. 3. Fig. 7 show the DO had a 3-moth lag with stratification, which cannot confirm the stronger the stratification, the lower DO (L 186). There must be some other mechanisms.

Special comments:

1. There was a significant relationship between hypoxia formation and estuarine dynamic process. But the paper does not verify the hydrodynamic process of the physical model. The circulation pattern shall be figured out.

2. L 79-83 the reference and web address for GDEM, SODA, NECP shall be listed.

3. L 87 both chlorophyll and chlorophyll-a are used in this MS, shall they be consistent?

4. Line 97-98: The initial values of phytoplankton, zooplankton and detritus concentrations were set to 0.5, 0.25 and 0.25 times of chlorophyll-a concentration, respectively. Why to set the initial values of phytoplankton, zooplankton and detritus concentrations were 0.5, 0.25 and 0.25 times of chlorophyll-a concentration? What is the basis? Need the reference to support.

5. L 108-109 the model's initial and open boundary are set by the GDEM, it is not a good idea to compare the model's result with the GDSM, at least the SST from RS can be the observed data to evaluate the model's performance.

6. L 108-116 there are serval upwelling systems in east china sea, can the model reproduce them?

7. L120 Changejiang estuary and Yangtze Estuary shall be consistent.

8. L133 it is mentioned the high nitrate is along section 123E, which is from Yangtze river. But in Lin 136-137, you mention it is Taiwan warm upwelling. Please make it clear.

9. 137-138 no evidence for the Taiwan warm upwelling (why can upwelling be warm?). The high nutrients at surface in fig. 3.

10. L 154, the basic condition shall be listed

11. Table 2 the value of Q, Qconst shall be listed. The unites for N2?

12. L180 the N2 is for the red rectangle, but the O2 is only for one point? N2 is vertical averaged?

13. Section 3.3, what is the mechanism of wind direction affect the DO? The circulation is changed in this case?

14. L 240 findings => results

15. S 4.1 what is the relation of Chlorophyll with DO? Higher river discharge will increase both N2 and Chlorophyll, both will lead to less DO at the bottom because of less DO mixing from surface and the decomposition of the organic in bottom layer. So which is more important?

―――――――――――――――――――